# Hydrophilic Interaction Chromatography Coupled with Charged Aerosol Detection for Simultaneous Quantitation of Carbohydrates, Polyols and Ions in Food and Beverages

**DOI:** 10.3390/molecules24234333

**Published:** 2019-11-27

**Authors:** Johannes Pitsch, Julian Weghuber

**Affiliations:** 1FFoQSI-Austrian Competence Centre for Feed and Food Quality, Safety & Innovation, Head Office, FFoQSI GmbH, Technopark 1C, 3430 Tulln, Austria; johannes.pitsch@ffoqsi.at; 2Center of Excellence—Food Technology and Nutrition, University of Applied Sciences Upper Austria, 4600 Wels, Austria

**Keywords:** hydrophilic interaction chromatography, charged aerosol detection, sugars, polyols, ions

## Abstract

Here, we report an accurate and versatile method for the simultaneous determination of 17 sugars (arabinose, erythrose, fructose, galactose, glucose, isomaltulose, lactose, lyxose, maltose, maltotriose, mannose, raffinose, rhamnose, ribose, sucrose, sorbose and xylose), seven polyols (erythritol, inositol, lactitol, maltitol, mannitol, sorbitol and xylitol), five ions (K^+^, Br^−^, Cl^−^, NO^3−^ and SO_4_^2−^) and the pseudosaccharide acarbose. For compound separation, hydrophilic interaction chromatography (HILIC) coupled to a corona charged aerosol detector (CAD) was used. The method was validated for linearity, precision, reproducibility, retention factor and optimal injection volume. Standards were measured in the range of 1–1000 mg L^−1^ and showed good intraday and interday repeatability, as well as precision (relative standard deviation (RSD) < 5%). The LODs and LOQs for the 30 analytes were in the range of 0.032–2.675 mg L^−1^ and 0.107–8.918 mg L^−1^, respectively. This method exhibited correlation coefficients of at least R^2^ > 0.97 for all analytes. The method was tested in 24 food and beverage samples to validate the separation efficiency and sensitivity in natural food matrices and to show the practicability of its use for routine food analysis.

## 1. Introduction

With the development of refractive index detectors (RIDs), carbohydrate analysis via high-performance liquid chromatography (HPLC) has become popular, enabling detection limits in the range of 0.34 ± 0.18 g L^−1^ [1]. HPLC-RID offers fast and simple analysis of carbohydrates, polyols and substances that change the refractive index of the solvent, such as metal cations and hydrocarbons [2,3,4]. However, the main disadvantage of RI detection is its restriction to isocratic methods. If the separation of monomers and polymers is required in one run, the use of HPLC-RID can take hours to complete [5]. UV/VIS detection by HPLC-DAD (diode array detector) offers similar detection limits as HPLC-RID but requires the derivatization of carbohydrates and most polyols pre- or postinjection onto the column [6]. As a result, sample preparation is more complex, but gradient elution can be implemented. Fluorescence and HPLC-FLD (fluorescence detector) is a technique that further enhances the sensitivity and selectivity of detection compared to those of HPLC-DAD [7]. However, HPLC-FLD also requires additional derivatization steps. The determination of sugars and polyols by gas chromatography coupled to mass spectrometry (GC-MS) also requires time-consuming derivatization steps and includes expensive internal standards rendering the procedure unsuitable for routine analysis [8]. Moreover, the LOD and RSD values in comparison to those of HPLC-RID have not proven to be advantageous in GC-MS detection. In general, methods that do not require derivatization steps are preferred, as derivatization itself might cause loss of analytes, indicated by RSD ratios in the range of 5.5–22.8% and recovery rates of 59.0–127.3% [9]. HPLC-MS/MS detection is used for the structural elucidation of unknown carbohydrate polymers. However, HPLC-MS/MS requires expensive analytical devices and sophisticated user knowledge [10]. Therefore, most laboratories are not able to use HPLC-MS/MS in routine carbohydrate analysis.

Introduced in 2002 [11], analytical methods for the detection of carbohydrates via charged aerosol detection (CAD) have become widely used in recent years [11,12,13,14,15,16,17,18]. In charged aerosol detection, analyte particles are produced by nebulizing the column eluent with nitrogen, followed by drying. Subsequently, analyte particles are charged by a high-voltage corona wire, whereas the number of charges is directly proportional to the particle size of the analyte. The amount of charges, measured by a highly sensitive electrometer, is directly proportional to the quantity of the analyte present. Several methods utilizing charged aerosol detection have been successfully published in the past [14,19,20,21,22]. Its use for carbohydrate analysis is still limited to a small number of studies as well as to a limited number of compounds that are detectable with those methods [14,23]. ELSD-, NMR- and MS-based techniques are well established [14,24,25] but require costly sample preparation or derivatization steps [26,27]. The method we propose here is fast, versatile and inexpensive compared to LC-MS or NMR and offers an efficient separation mechanism, thereby increasing the number of detectable compounds to an extent, which, to the best of our knowledge, has not yet been published.

Carbohydrates are essential for energy production in the human body, especially as monomers such as glucose and fructose. Over the last few decades, studies have indicated that the emergence of diabetes mellitus type 2 is closely related to various parameters, including the consumption of both glucose and fructose [28,29,30]. Furthermore, sugar alcohols have a negative influence on people suffering from irritable bowel syndrome (IBS) [31,32]. Therefore, it is essential to gain insight into the carbohydrate composition of relevant food and beverages and thereby mitigate the consumption of foodstuff of unknown carbohydrate and sugar alcohol composition.

Carbohydrate analysis is prescribed by EU law so that customers can assess the nutritional value of foodstuff. Sugar alcohols are reduced forms of their corresponding carbohydrates and are known for their laxative effect and, as mentioned before, their influence on IBS [33]. Some of them, such as sorbitol, mannitol, isomalt, maltitol, lactitol, xylitol and erythritol, are used as nutritive sweeteners. Therefore, the content of carbohydrates and sugar alcohols (polyols) must be continuously monitored by manufacturers and legislation agencies to ensure customer safety and information regarding the potential health concerns of certain ingredients.

In addition to carbohydrates and polyols, monovalent ions, such as K^+^ and Cl^−^, are long-known to play essential roles in carbohydrate uptake [34]. Increased K^+^ uptake is linked to beneficial effects on cardiovascular disease, coronary heart disease and type 2 diabetes [35,36]. Regardless of the significance of carbohydrate analysis, various inorganic ions in food and beverages generally need to be quantitated, including i) SO_3_^2−^, which is added to foodstuff as an antioxidative agent and oxidized to SO_4_^2−^ during storage, ii) Cl^−^ for determining table salt content and iii) NO_3_^−^ maximum concentrations in drinking water are regulated by EU laws. Typically, these ions need to be measured separately from carbohydrates by suitable technologies such as ion chromatography (IC) [37]. Therefore, new methodologies for the parallel quantitation of ions and carbohydrates without the need for additional sample preparation steps is of great interest.

Based on the abovementioned restrictions of currently available methods and the importance of food analysis, especially for sugars and sugar alcohols, there is a great demand for fast and simple analytical methods, which offer robustness, reliability and reproducibility and can be applied to as many sample matrices as possible to cover a broad range of foodstuff and beverages. The aim of this study was to develop and validate a new chromatographic method for HILIC-CAD, enabling the simultaneous detection of 17 sugars (arabinose, erythrose, fructose, galactose, glucose, isomaltulose, lactose, lyxose, maltose, maltotriose, mannose, raffinose, rhamnose, ribose, sucrose, sorbose and xylose), seven sugar alcohols (erythritol, inositol, lactitol, maltitol, mannitol, sorbitol and xylitol), five ions (K^+^, Br^−^, Cl^−^, NO_3_^−^ and SO_4_^2−^) and the pseudo saccharide acarbose. The procedure was applied for the determination of these 30 compounds in 24 beverage and food samples with very good results.

## 2. Results and Discussion

In this study, we present a new HILIC-CAD method to separate a total of 30 different analytes, including 17 sugars, seven sugar alcohols, five ions and one pseudo saccharide. Analyte separation was achieved in a single run with minimal sample pretreatment.

### 2.1. Optimization of HILIC-CAD Conditions

The optimum HILIC-CAD conditions were chosen in terms of peak symmetry, equilibration time, chromatographic analysis time, resolution and retention factor (k). The resolution of the investigated compounds was tested by changing the ratio of acetonitrile and water, both in isocratic and gradient runs. The application of buffer gradients was examined in isocratic runs. The influence of pH was tested regarding retention factor (k), peak shape and resolution at values of 4.75, 7.00, 8.25, 9.25 and 11.00 (without ammonium acetate, 0.1% ammonia). Under isocratic conditions, the influence of column temperature at 25, 30, 35, 40, 50 and 60 °C on the parameters mentioned above was also evaluated.

#### 2.1.1. Optimization of Mobile Phase in Isocratic Elution

Isocratic runs were examined for ACN:H_2_O ratios of 90:10, 87.5:12.5, 85:15, 82.5:17.5, 80:20, 75:25, 70:30 at flow rates from 0.2 to 0.6 mL min^−1^. At an ACN:H_2_O ratio of 80:20, the influence of pH value and ammonium acetate buffer at 2.5, 5.0, 7.5, 10, 20 and 40 mmol L^−1^ was tested. For mobile phase selection, methanol, ethanol, isopropanol, acetone and ACN were tested under isocratic conditions with an 80:20 ratio of organic solvent:H_2_O. ACN was chosen as the organic solvent for gradient run evaluations based on mobile phase selection tests and the recommendation of the column manufacturer. In accordance with Hutchinson et al. [14], carbohydrates and polyols showed the best separation efficiency and retention factors (k) with ACN. Furthermore, the CAD revealed that ionization with ACN achieved the lowest baseline noise compared to that of all tested organic solvents. However, it was impossible to obtain satisfactory separation in isocratic separation, especially with regard to lactose, lactitol, maltose, maltitol, galactose and glucose. It can be concluded that the increase of water content in the mobile phase shortens retention times but decreases resolution. At a 70:30 ratio of ACN:H_2_O, the coelution of monomers was inevitable. A solvent ratio of 90:10 (ACN:H_2_O) prolonged retention, particularly for acarbose, inositol, isomaltulose, lactitol, lactose, maltitol, maltose, maltotriose, raffinose and sucrose, resulting in poor peak shape due to peak broadening. At an 80:20 ratio of ACN:H_2_O and 0.3 mL L^−1^, separation of monomers was impossible for glucose, galactose, xylitol and xylose but sufficient for polymers >1 monomeric unit. As the 80:20 ratio (ACN:H_2_O) showed the best overall resolution of the seven tested concentrations, it was utilized for subsequent evaluation of varying pH values and buffer concentrations.

#### 2.1.2. Evaluation of the Buffer Gradient Conditions

The application of buffer gradients showed irreproducible retention factors for polymers of *n* > 1 and was therefore not taken into consideration for further testing. Isocratic ammonium buffer concentrations lower than 7.5 mmol L^−1^ reduced equilibration time and reproducibility, and concentrations above 10 mmol L^−1^ did not improve separation efficiency but led to increased baseline drift. A concentration of 10 mmol L^−1^ was therefore considered the best compromise between reproducibility and baseline drift.

#### 2.1.3. pH Value Optimization

The sole use of NH_3_ for pH adjustment, instead of the combination NH_3_ and ammonium acetate as buffer, seemed appropriate because NH_3_ is more readily evaporated in the CAD. When using NH_3_ for pH adjustment, the retention factor for carbohydrates and polyols decreased, thus leading to earlier elution and decreased resolution. The same effect was observed with detection of dissolved ions, respectively. Ions showed almost no retention and immediate elution with retention factor k < 1. The importance of ion detection must be emphasized when food samples are analyzed. Therefore, ammonia as the sole buffer component seemed inadequate. A pH <8.25 at 35 °C showed undesired anomer separation between the α- and β-ring structures of carbohydrate monomers and was therefore excluded from further testing. With maximum solvent buffer capacity (ammonium acetate pK_a_ = 9.25), loss of resolution and heavy baseline increase were observed. It can be deduced that this circumstance is caused by the CAD working principle: in the ion trap compartment, not only the sample but also the buffer particles become positively charged. At the highest buffer capacity, the number of applied charges increases dramatically, causing a high workload for the detector’s ion trap. This leads to poor response and a severe decrease in chromatographic resolution. Ideal mobile phase buffer composition was identified to be at pH values near the lower limit of the buffer salt pK_a_, in the case of ammonium acetate, this corresponded to pH = 8.25.

#### 2.1.4. Equilibration Time

The equilibration time was strongly dependent on the duration of the initial eluent concentration in the column and less on the total flow rate. Higher flow rates did not decrease equilibration time significantly. According to the column manufacturer’s specifications, 8–10 column volumes of initial buffer concentration for re-equilibration of the column are recommended. For the 2.1 mm column used, equilibration times >30 min proved to be sufficient for reproducible retention factors (k).

#### 2.1.5. Column Temperature

The impact of column temperature was evaluated at an 80:20 ACN:H_2_O ratio, pH = 8.25 and a 10 mmol L^−1^ buffer concentration. Temperatures <30 °C resulted in impaired peak shape, high column backpressure and insufficient resolution, most likely due to the lower interaction of analytes present in the mobile phase with the surface water layer and ion exchange groups of the column. At 25 °C, maltitol and maltose separation was impossible, and galactose showed peak symmetry values <0.8. Overall, the signal intensity revealed an increase with temperature but a decrease in resolution. Separation of glucose and galactose was impossible at >40 °C. The signal-to-noise ratios as well as the peak heights at 40 °C showed the best results for all six tested temperatures and was defined as the optimum temperature parameter for method optimization.

#### 2.1.6. Sample Solvent Optimization

According to the manufacturer’s recommendations, matching organic solvent content in eluent and sample diluent can improve the peak shape of analytes and separation efficiency. The solubility of analytes was therefore tested at varying ACN concentrations. Dilution of 10 mg of glucose in 1 mL of 100% (*v*/*v*) ACN resulted in an insoluble residue, whereas dilution in 85% (*v*/*v*) ACN showed fewer residual particles, and in 60% (*v*/*v*) ACN, glucose was dissolved. All three obtained solutions were injected after centrifugation. The results clearly showed that glucose was lost with increasing ACN content. No loss of analytes was observed when dissolving the residue of all samples in 60% (*v*/*v*) ACN and centrifuging prior to subsequent chromatographic analysis.

These optimized parameters resulted in negligible baseline noise and enhanced peak symmetry, resolution and retention factors as well as reduced equilibration and analysis times. Therefore, the method allowed for quantitative determination of the target analytes. Chromatographic peaks were identified by comparing their retention times with those of reference compounds. A representative chromatogram for a mixture of 30 different analytes is shown in Figure 1.

### 2.2. Validation of the Method

The analytical method was fully validated by evaluating the nonlinear calibration range, repeatability, correlation coefficient, interday precision, intraday precision, relative standard deviation (RSD), limit of quantitation (LOQ), limit of detection (LOD) and retention factor (k). The data concerning method validation are summarized in Table 1 and Table 2.

#### 2.2.1. Linearity and Detector Response

Due to the nature of charged aerosol detection, the validity of linear calibration is questionable. Although linear calibration can deliver satisfying square correlation coefficients of the calibration curve [23], we aimed for a different approach. Standards were diluted and measured within their calibration range with linear calibration and compared to nonlinear calibration. Nonlinear regression showed higher accuracy in test standards and was therefore selected as the calibration method of choice. Nonlinear calibration curves were obtained by injecting five working solutions five times ranging from 10.0 to 1000 mg L^−1^ for carbohydrates and polyols and 1.0 to 100 mg L^−1^ for ions, respectively. Although smaller working ranges could potentially yield a near-linear calibration curve [11,38], more flexibility in routine analysis was achieved by using working ranges covering four orders of magnitude for carbohydrates and polyols. Responses obtained in the examined range could be expressed by a quadratic equation f(x) = a + bx + cx^2^ with good correlation coefficients (R^2^) ≥ 0.999 for 28 of the 30 analytes (Table 1). However, NO_3_^−^ and Cl^−^ constitute an exception, as the working range over three orders of magnitude was found to be linear. The above-mentioned lower concentrations for ion standards were chosen according to the expected concentrations in the samples and to avoid column overload.

#### 2.2.2. Precision

Repeatability was calculated for each substance to evaluate the precision of the method. The precision was expressed as relative standard deviation (RSD% = 100 × SD/mean). The obtained retention times and peak areas are indicated in Table 1. Repeatability and reproducibility were obtained by the consecutive injection of five working standards in different concentrations for each substance (intraday precision) followed by the same measurement after five and ten days (interday precision). The observed RSD ranged from 0.01 to 4.78% (Table 1). The precision of the method can therefore be considered very high, as the relative standard deviations for the retention times as well as the peak heights were below 5%.

#### 2.2.3. Detection and Quantitation Limits

The estimation of the detection and quantitation limits was carried out according to the international council for harmonization and technical requirement (ICH) guidelines [39]. Based on the signal-to-noise ratio, LOD and LOQ are defined as 3:1 and 10:1, respectively. The LOD and LOQ values determined for all target analytes are given in Table 2. LOD and LOQ were in the range of 0.247–2.675 mg L^−1^ and 0.885–8.918 mg L^−1^ for carbohydrates and polyols and 0.032–0.208 mg L^−1^ and 0.107–0.693 mg L^−1^ for ions, respectively. Therefore, this approach resulted in lower LOD and LOQ limits in comparison to those of RI and UV/VIS detection, as previously reported [6].

#### 2.2.4. Retention Factor (k)

The interaction of the analyte molecule with the stationary phase of the analytical column can be described by the retention factor k. If a compound is coeluted with the injection peak, the retention factor is equal to 0. Factors ≥2 are required by ICH to prove that the analyte is actually retained by stationary phase interaction [39]. Here, we report on retention factors within the range of 0.9–36.8, with NO_3_^-^ being eluted first (Table 1). Based on ICH recommendations, NO_3_^-^ should be considered to be excluded from quantitation. However, considering a good R^2^ value >0.999 and reproducible intra- and interday retention times of 1.381 ± 0.007 min, the compound was further quantitated in subsequent experiments.

#### 2.2.5. Injection Volume

Column overload due to high analyte concentration in the sample can lead to insufficient baseline separation of proximate analytes. Furthermore, in the case of overall sample overload, a useful strategy is the reduction of the injection volume. The same mixture of 15 analytes was consecutively injected five times, and the area under the curve was quantitated (Figure 2). A change in injection volume showed a linear correlation with the peak height with R^2^ > 0.99 for all analytes.

Although all 15 compounds show good linear correlation, the change of injection volume required recalibration of the method. This finding is supported by the observation that the slope for each substance is different due to a nonlinear detector response that seems to be dependent on the analyte. The slope of sorbitol (0.89) was highest, whereas the slope of erythritol (0.27) was the lowest of all 15 analytes.

Therefore, it is possible to adapt the injection volume within the range of 1–5 µL to meet the analytical demands without losing precision in quantitation.

### 2.3. Quantitative Analysis of Food Samples

The method described here was applied for the determination of the 30 target analytes in various beverage and food samples. Coelution of NH_4_^+^, abundant in the buffer solution, with Na^+^ inhibits quantitation of sodium in the food samples. Thus, sodium was not included in the quantitation. Chromatograms of three samples with different analyte complexity are shown in Figure 3. The concentrations found for all analytes are given in Table 3. Fructose was the most abundant compound in the samples tested. None of the samples contained all of the target analytes. RSD values were calculated for six repeated injections, two subsequent injections of each sample immediately after preparation, and after five and ten days. The values of all peaks per sample were averaged for each injection and day. The obtained RSD values were <5% and demonstrated excellent sample stability and repeatability. Those values, despite large variations between the sample constituents and matrices, underline the applicability of this method for routine analysis.

After ten days of sample storage in the autosampler at 25 °C, no neither precipitation nor discoloration, which could have caused loss of analytes, were visible in any sample. Fourteen of the 24 tested beverage and food samples contained nutritional information on their packaging. For the determination of the carbohydrate content of a product, commercial vendors offer standard methods based on density, redox potential or enzymatic tests. Therefore, we compared the results obtained from our method with the specifications given on the product labels. The obtained results matched (100 ± 0%) with labeled values for Red Bull and Coca Cola. The highest deviation was visible in the non-alcoholic beverage sample Zipfer Hell (172%). Other labeled samples showed 99 ± 11.4% matching results. The high deviation between the reported Zipfer Hell contents with our measured values might be caused by the use of an unsuitable analytical method by the manufacturer. The small deviations detected for some samples might be explained by different analytical approaches or methods used by food manufacturers, which are not optimized for the respective food matrix. For example, reducing agents in the product can alter the value obtained by redox titration. Furthermore, strong coloration of the product can impair spectrometric detection. By detection principle, our method was not influenced by the mentioned parameters.

With respect to the quantitated ions, there was no sample with all ions present. As the most abundant ion, the potassium content in Heinz tomato ketchup was the highest of all samples and correlated with the potassium content in tomatoes published by independent institutions [40]. The increased sulfate content of both wine samples was presumably caused by the addition of sulfite as an antioxidizing agent during the production process. Similar to the coelution of Na^+^ and NH_4_^+^, the coelution of SO_4_^2−^ and SO_3_^2−^ is likely, although the oxidation of sulfite cannot be excluded.

The simultaneous detection of the present ions can be stated as an additional benefit of this method. Table salt (NaCl) content is determined in the food industry by refractometry, electrical conductivity, titration or ion-selective electrodes [41]. These methods are frequently susceptible to over- and/or underestimation of table salt content. With the method proposed here, simultaneous and selective detection of anions and cations is possible, with further potential to reduce costs for food manufacturers.

## 3. Materials and Methods

### 3.1. Chemicals, Food Samples and Standards

LC-MS-grade acetonitrile and ammonium acetate, as well as NH_3_, were obtained from VWR (Vienna, Austria). Analytical-grade acarbose, arabinose, erythritol, fructose, galactose, glucose, inositol, lactitol, lactose, maltitol, raffinose, rhamnose, ribose, sucrose, xylitol, potassium chloride, ammonium iodide and potassium bromide were purchased from Sigma Aldrich (Schnelldorf, Germany). Erythrose, isomaltulose, lyxose, maltose, maltotriose, mannitol, mannose, sorbitol, sorbose, xylose, sodium nitrate and sodium sulfate were purchased from VWR (Vienna, Austria). LC-MS-grade water (< 0.055 µS cm^−1^) from an ultrapure water purification system (Sartorius, Göttingen, Germany) was used for both elution and sample preparation. Food samples produced by various companies were obtained from local supermarkets. Sample matrices have been selected over a wide range of beverages and food to investigate the impact on sample preparation and robustness of the analytical method. Food and beverages were stored at the recommended temperature until analysis.

### 3.2. Sample and Standard Preparation

Samples were diluted as follows: beverage samples, 1:500; yogurt/milk samples, 1:100 and alcoholic beverages, 1:10. Samples containing gas were degassed in an ultrasonic bath prior to dilution. All samples were diluted in 60% (*v*/*v*) ACN and centrifuged at 17,000× *g* at 25 °C for 5 min. After centrifugation, 800 µL of supernatant was used for analysis. The analytical standards were diluted in water prior to final dilution in 60% (*v*/*v*) ACN and centrifuged at 17,000× *g* at 25 °C for 5 min. Carbohydrate and polyol standards were prepared at concentrations of 0.010, 0.050, 0.10, 0.50 and 1.0 g∙L^−1^, and ion standards were prepared at 0.001, 0.005, 0.010, 0.050, 0.100 g∙L^−1^.

### 3.3. Instrumentation

Chromatographic experiments were performed on an UltiMate 3000 UHPLC system (Thermo Fisher Scientific, Waltham, Massachusetts, MA, USA) equipped with a solvent degasser, quaternary pump, autosampler and thermostatic column compartment and coupled to a Corona Veo instrument (Thermo Fisher Scientific, Massachusetts, MA, USA). Data processing was carried out with Chromeleon 7.2.8 software (Thermo Fisher Scientific, Massachusetts, MA, USA), and the compressed air gas flow rate was automatically regulated and monitored by the CAD device. Data collection was set to 2.0 Hz at a filter constant of 3.6 s. The power function for response and signal correction was set to 1.30.

### 3.4. Chromatographic Conditions

Chromatographic separation was achieved using a WATERS Acquity UPLC BEH amide column (130 Å, 1.7 µm, 2.1 mm × 150 mm) maintained at 40 °C. The autosampler was kept at 25 °C. A guard column (Acquity UPLC BEH Amide VanGuard precolumn, Milford, MA, USA) was used to protect the column from particles. Mobile phase A was 85% ACN, and mobile phase B was 60% ACN. All mobile phases contained 10 mM NH_4_Ac adjusted to pH = 8.25 with NH_4_OH (25%, NH_3_ basis). The buffer components were filtered through a 0.2 µm membrane filter after preparation. The gradient program used is shown in Table 4. The autosampler needle was washed before and after each injection with 100 µL (20 µL s^−1^) of mobile phase A to prevent sample carryover between runs. The injection volume was 5 µL.

### 3.5. Statistics

Statistical analysis, i.e., linear regression and intra- and interday repeatability, was performed with Chromeleon 7.2.8 software and Microsoft Office Excel 365 (Redmond, Washington, WA, USA).

## 4. Conclusions

The method described in this study proved to be reproducible and precise, providing LODs and LOQs suitable for quantitation of various carbohydrates, polyols and ions in 24 food samples. Applicability in routine analysis along with good adaptability to a wide range of sample types was demonstrated.

Erythritol was not detected in any sample, rendering it most suitable for use as an internal standard. By the addition of erythritol, it was possible to deduce potential retention time shifts or deviances in detector response without interrupting the analysis. Although the mentioned problems did not occur during method development, the use of an internal standard is advisable. Coelution of both Na^+^/NH_4_^+^ and SO_4_^2−^/SO_3_^2−^ proved to be a downside of the method. In the future, the coelution of Na^+^/NH_4_^+^ could possibly be avoided by the use of a different buffer agent. Matching the sample diluent concentration to the final rather than the initial gradient composition prevented analyte loss as described in the section regarding sample solvent optimization. In conclusion, the application of HILIC chromatography and charged aerosol detection, in combination with versatile sample preparation, was demonstrated to be a reliable tool for routine food and beverage analysis.

Separation of carbohydrates is a difficult task as chemical similarity demands high column separation efficiency and prolonged method runtimes. To achieve the best compromise between method speed and reliability, the approach of quantitation by peak height instead of peak area was chosen and led to good results.

Sample matrix interference could further be lowered by increasing ACN concentration in the sample diluent or by the use of solid phase sample cleanup prior to analysis. Using 60% ACN showed some matrix interference, especially at the complex beer samples, but did not impair quantitation results.

Beer samples contained malto-oligomers with a degree of polymerization >3. Due to a lack of standards, it was not possible to quantitate those compounds.

## Figures and Tables

**Figure 1 molecules-24-04333-f001:**
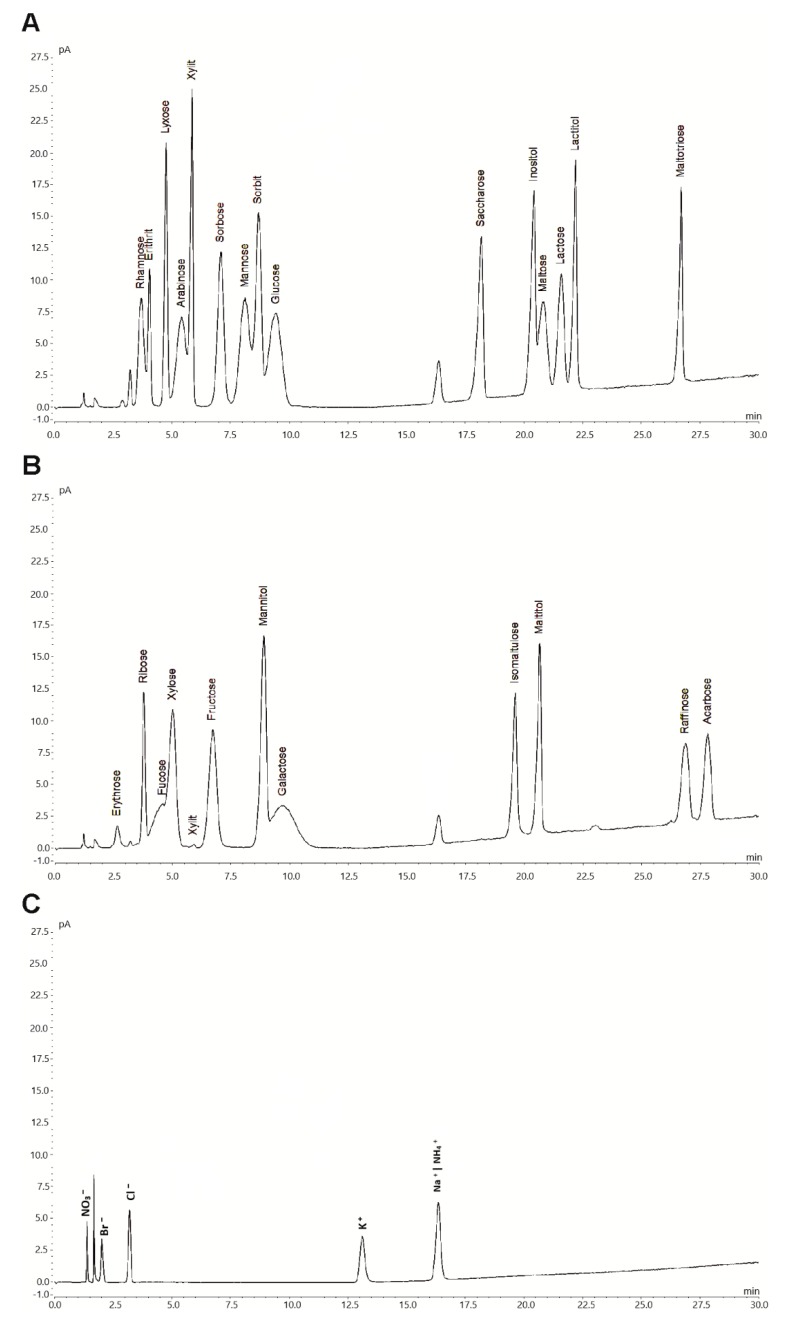
Hydrophilic interaction chromatography (HILIC) coupled with a corona veo SD charged aerosol detector (CAD) (HILIC-CAD) chromatogram of a standard mixture containing 17 different sugars (0.1 g L^−1^), 7 polyols (0.1 g L^−1^), 5 ions (0.01 g L^−1^) and acarbose (0.1 g L^−1^). For a clear representation, depicted chromatograms were divided for sugars and polyols (**A** and **B**), as well as for ions (**C**).

**Figure 2 molecules-24-04333-f002:**
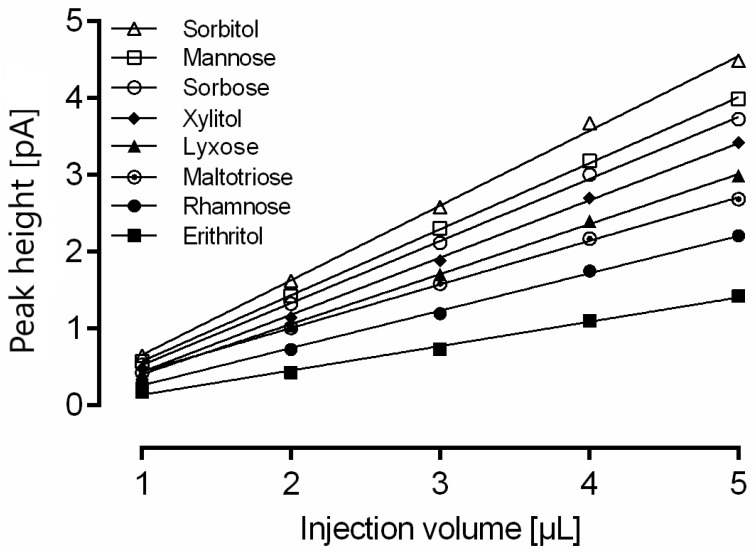
Impact of injection volume on calculated amount of rhamnose, erythritol, lyxose, xylitol, sorbose, mannose, sorbitol and maltotriose measured with optimized gradient conditions. For a clear representation, linear regression is only shown for eight representative analytes. Data are the average of five determinations.

**Figure 3 molecules-24-04333-f003:**
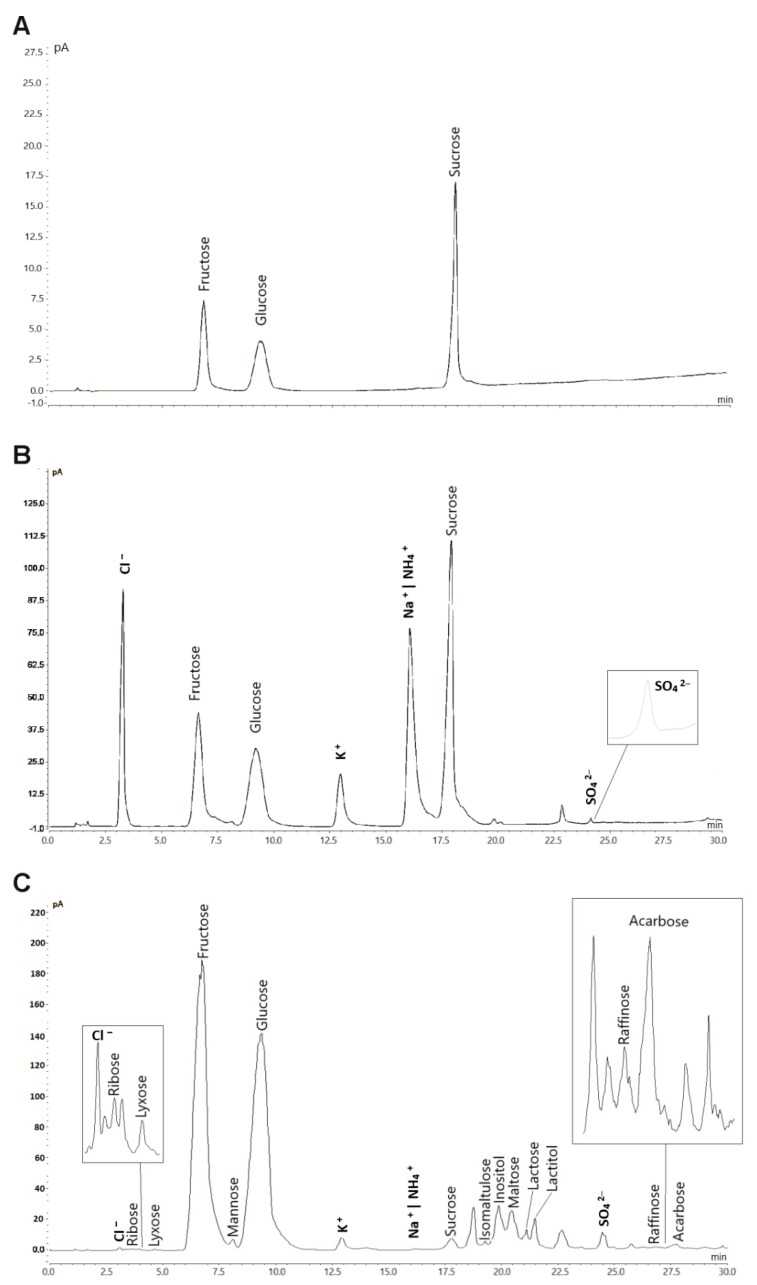
HILIC-CAD chromatograms of selected analyzed samples with different matrices and analyte complexity. (**A**) Coca Cola, (**B**) Felix hot ketchup and (**C**) Honey. The injection volume of the sample solution was 5 µL. Analysis was carried out with optimized gradient conditions.

**Table 1 molecules-24-04333-t001:** Validation data of the analytical methodology including calibration range, correlation coefficients, retention time as well as intra- and inter-day precision of retention time and RSD peak height. RSD values are based on the mean of indicated standard concentrations. Peak height was used for generation of calibration curves.

Substance	Calibration Curve Range [mg L^−1^]	R^2^	Calibration Curve	Mean RT [min]	RSD RT Intra-Day [%]	RSD RT Inter-Day [%]	RSD Peak Height Intra-Day [%]	RSD Peak Height Inter-Day [%]
**Sugars/polyols**								
Acarbose	10–1000	0.9999	y = 25.544x − 5.014x^2^	27.67	0.08	0.15	0.64	4.78
Arabinose	10–1000	0.9999	y = 30.616x − 3.624x^2^	5.33	0.18	0.25	0.05	4.35
Erythritol	10–1000	0.9997	y = 16.664x + 1.725x^2^	4.02	0.17	0.34	0.37	4.14
Erythrose	10–1000	0.9999	y = 3.163x + 0.825x^2^	2.64	0.48	0.50	0.85	1.96
Fructose	10–1000	0.9999	y = 37.475x − 6.623x^2^	6.66	0.26	0.42	0.25	1.10
Galactose	10–1000	0.9999	y = 34.953x + 1.197x^2^	9.78	0.41	0.32	0.13	2.54
Glucose	10–1000	0.9999	y = 43.235x − 8.471x^2^	9.29	0.34	0.32	0.60	2.49
Inositol	10–1000	0.9999	y = 359.115x − 668.680x^2^	20.44	0.27	0.30	1.06	2.45
Isomaltulose	10–1000	0.9999	y = 26.901x − 5.677x^2^	19.43	0.09	0.22	1.51	1.96
Lactitol	10–1000	0.9993	y = 29.872x − 7.346x^2^	22.08	0.14	0.17	0.16	2.07
Lactose	10–1000	0.9997	y = 27.227x − 5.899x^2^	21.47	0.08	0.14	0.13	0.95
Lyxose	10–1000	0.9750	y = 30.117x − 10.489x^2^	4.69	0.15	0.15	0.29	3.40
Maltitol	10–1000	0.9999	y = 31.769x − 10.072x^2^	20.63	0.09	0.09	2.72	3.27
Maltose	10–1000	0.9889	y = 26.819x − 8.189x^2^	20.79	0.10	0.19	0.90	1.29
Maltotriose	10–1000	0.9999	y = 25.737x − 5.700x^2^	31.16	0.05	0.13	0.38	1.27
Mannitol	10–1000	0.9915	y = 45.186x − 11.264x^2^	8.92	0.34	0.46	1.15	1.38
Mannose	10–1000	0.9999	y = 34.622x + 47.409x^2^	7.97	0.10	0.45	0.46	3.36
Raffinose	10–1000	0.9999	y = 26.835x − 5.413x^2^	26.87	0.09	0.13	4.33	4.94
Rhamnose	10–1000	0.9999	y = 24.266x − 1.073x^2^	3.57	0.01	0.01	0.45	1.96
Ribose	10–1000	0.9999	y = 18.108x − 1.428x^2^	3.69	0.25	0.39	0.59	4.43
Saccharose	10–1000	0.9999	y = 32.742x − 7.018x^2^	18.07	0.12	0.26	0.18	0.22
Sorbitol	10–1000	0.9999	y = 44.572x + 10.392x^2^	8.54	0.38	0.47	2.93	2.96
Sorbose	10–1000	0.9999	y = 36.326x − 7.002x^2^	7.05	0.24	0.31	1.58	3.00
Xylitol	10–1000	0.9997	y = 32.644x − 6.659x^2^	5.77	0.08	0.36	3.09	3.85
Xylose	10–1000	0.9999	y = 45.659x − 7.009x^2^	4.95	0.10	0.35	0.54	1.61
**Ions**								
Cl^-^	1–100	0.9999	y = 0.085x	3.19	0.67	0.83	0.36	2.97
Br^-^	1–100	0.9999	y = 44.404x − 58.056x^2^	2.01	0.48	0.56	0.29	1.80
NO_3_^-^	1–100	0.9999	y = 0.033x	1.38	0.33	0.51	0.42	4.26
SO_4_^2-^	1–100	0.9998	y = 17.800x − 17.769x^2^	24.40	0.09	0.15	0.02	1.29
K^+^	1–100	0.9998	y = 142.061x − 92.797x^2^	13.14	0.05	0.13	0.05	4.74

**Table 2 molecules-24-04333-t002:** LOD (S:N = 3:1) and LOQ (S:N = 10:1) values and retention factors (k) for 30 analytes (*n* = 5); system volume dead time = 0.721 min; injection volume = 5 µL; optimized gradient conditions.

Substance	LOD [mg L^−1^]	LOQ [mg L^−1^]	k
**Sugars/polyols**			
Acarbose	0.51	1.71	36.8
Arabinose	1.00	3.35	6.4
Erythritol	0.65	2.16	4.5
Erythrose	2.68	8.92	2.6
Fructose	0.45	1.51	8.1
Galactose	1.83	6.10	12.2
Glucose	0.81	2.69	11.8
Inositol	0.27	0.90	26.8
Isomaltulose	0.39	1.30	25.7
Lactitol	0.25	0.82	29.2
Lactose	0.36	1.18	28.4
Lyxose	0.29	0.98	5.5
Maltitol	0.27	0.88	27.1
Maltose	0.49	1.65	27.3
Maltotriose	0.46	1.54	35.3
Mannitol	0.31	1.03	10.8
Mannose	0.64	2.13	10.0
Raffinose	0.27	0.89	35.5
Rhamnose	0.79	2.62	4.0
Ribose	0.51	1.71	4.2
Saccharose	0.29	0.98	23.7
Sorbitol	0.34	1.15	11.1
Sorbose	0.46	1.55	8.6
Xylitol	0.30	1.01	6.9
Xylose	0.63	2.09	5.9
**Ions**			
Cl^-^	0.07	0.22	3.4
Br^-^	0.08	0.26	1.7
NO_3_^-^	0.06	0.19	0.9
SO_4_^2-^	0.16	0.54	32.3
K^+^	0.21	0.69	16.9

**Table 3 molecules-24-04333-t003:** Concentration of sugars, polyols and ions in food samples and beverages under study.

Product	H_2_O	Red Bull	Red Bull Sugarfree	Zipfer Hell	Erdinger Weißbier	Stiegl Columbus	Stiegl	Eggenberg Urbock	Eggenber Hopfenspiel	Grieskirchner Pils	Blue Zweigelt	Green Veltliner	Coffee Normal	Coffee Strong	Smoothie	Hakuma	Milk	Fuze Tea	Cappy Antiox.	Hohes C	Apple Juice	Ketchup	Yoghurt	Honey	Coca Cola
Sugars/polyols	[g L^−1^]																								
Acarbose	-	-	-	0.16	0.49	0.31	0.76	1.42	0.21	0.52	0.03	0.04	0.07	0.07	-	-	-	-	-	-	-	-	-	12.45	-
Arabinose	-	-	-	3.19	0.17	0.20	0.22	0.16	0.26	0.25	0.18	0.24	-	-	-	-	-	-	-	-	-	-	-	-	-
Erythritol	-	-	-	-	-	-	-	-	-	-	-	-	-	-	-	-	-	-	-	-	-	-	-	-	-
Erythrose	-	-	-	-	0.11	-	0.15	-	-	0.11	-	-	0.30	0.29	-	-	-	-	-	-	-	-	-	-	-
Fructose	-	20.50	-	19.75	0.12	0.18	0.22	-	0.05	0.18	1.93	0.63	0.37	0.38	70.96	82.49		7.11	48.78	63.85	68.58	31.86	-	369.29	22.74
Galactose	-	-	1.26	-	-	-	-	-	-	-	-	-	-	-	-	-	-	-	-	-	-	-	12.53	-	-
Glucose	-	55.30	-	39.09	-	-	-	-	-	-	0.27	0.38	-	-	28.81	7.51	-	12.80	23.88	24.54	19.30	32.49	-	271.46	21.40
Inositol	-	-	-	-	-	-	-	-	-	-	-	-	-	-	-	-	-	-	-	-	-	-	-	37.00	-
Isomaltulose	-	-	-	-	-	-	-	-	-	-	-	-	-	-	-	-	-	-	-	-	-	-	-	13.45	-
Lactitol	-	-	-	-	-	-	-	-	-	-	0.55	0.07	-	-	-	-	-	-	-	-	-	-	-	6.09	-
Lactose	-	-	-	9.39	0.15	0.31	0.69	1.34	-	0.26	-	-	-	-	-	-	57.54	-	-	-	-	-	38.47	6.72	-
Lyxose	-	-	-	5.85	0.28	0.32	0.33	-	-	0.37	0.36	0.27	0.25	0.25	-	-	-	-	-	-	-	-	-	3.45	-
Maltitol	-	-	-	-	-	2.88	-	-	-	-	0.05	0.09	-	0.04	-	-	-	-	-	-	-	-	-	-	-
Maltose	-	-	-	26.20	-	-	-	26.42	-	-	0.07	0.02	-	-	-	-	-	-	-	-	-	-	-	44.50	-
Maltotriose	-	-	-	11.78	0.15	4.51	3.04	12.66	0.08	1.61	-	-	-	-	-	-	-	-	-	-	-	-	-	-	-
Mannitol	-	-	-	-	0.02	-	0.10	-	-	0.02	0.22	0.01	-	-	-	-	-	-	-	-	-	-	-	-	-
Mannose	-	-	-	0.09	0.05	0.06	0.08	0.24	0.03	0.07	1.23	1.52	0.67	0.76	-	-	-	-	-	-	-	-	-	16.27	-
Raffinose	-	-	-	0.05	0.33	0.50	0.80	1.47	-	0.51	-	-	-	-	-	-	-	-	-	-	-	-	-	4.76	-
Rhamnose	-	-	3.02	-	-	-	0.34	-	-	0.32	0.56	-	-	-	-	-	-	-	-	-	-	-	-	-	-
Ribose	-	-	-	5.78	0.27	0.33	-	0.41	0.19	-	-	0.56	-	-	-	-	-	-	-	-	-	-	-	6.95	-
Sucrose	-	34.40	-	-	-	-	-	-	-	-	-	-	0.01	-	26.83	-	-	27.67	16.17	29.55	17.47	105.12	-	41.20	61.79
Sorbitol	-	-	-	-	0.07	0.06	0.13	0.20	0.02	0.07	0.21	0.06	-	-	-	-	-	-	1.16	1.75	1.63	-	-	-	-
Sorbose	-	-	-	8.88	0.53	0.46	0.62	1.07	0.23	0.48	-	-	-	-	-	-	-	-	-	-	-	-	-	-	-
Xylitol	-	-	-	0.03	0.07	0.08	0.07	0.18	0.03	0.07	0.11	0.08	0.19	0.20	-	-	-	-	-	-	-	-	-	-	-
Xylose	-	-	-	-	-	-	-	-	0.04	-	-	-	-	-	-	-	-	-	-	-	-	-	-	-	-
**Ions**	**[mg L^−1^]**																								
Br^–^	-	-	-	-	-	22	22	32	-	-	-	10	149	146	-	-	-	-	-	-	-	-	-	-	-
Cl^–^	-	-	-	180	420	191	215	285	146	228	-	-	31	-	-	-	650	-	72	-	-	2461	558	686	-
SO_4_^2–^	-	201	-	91	207	295	223	213	120	265	1602	999	121	128	-	-	-	-	-	-	-	48	-	238	-
NO_3_^–^	-	-	1	-	-	-	-	-	-	-	-	-	2	2	-	-	-	-	-	-	-	-	-	-	-
K^+^	-	-	-	291	516	545	512	841	305	504	752	791	2048	1996	657	-	1043	-	502	597	367	11415	1085	6159	-

**Table 4 molecules-24-04333-t004:** Solvent gradient elution program used for the elution of sugars, polyols and ions. Negative values represent equilibration times.

Time [min]	%A	%B	Flow Rate [mL min^−1^]
−35.000	100.00	0.000	0.300
−30.100	100.00	0.000	0.300
−30.000	100.00	0.000	0.150
−11.000	100.00	0.000	0.150
−10.000	100.00	0.000	0.300
0.000	100.00	0.000	0.300
10.000	100.00	0.000	0.300
35.000	0.000	100.00	0.300
45.000	0.000	100.00	0.300

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
