# Peer review of "Hydrophilic Interaction Chromatography Coupled with Charged Aerosol Detection for Simultaneous Quantitation of Carbohydrates, Polyols and Ions in Food and Beverages"

_molecules, 2019, doi:10.3390/molecules24234333_

Round 1

Reviewer 1 Report

Reviewer report on manuscript Molecules-645595

The submitted research presents the development and application of a HILIC-CAD method of the simultaneous determination of carbohydrates, polyols, and ions in foodstuffs.

After careful research in the electronic databases (Scopus, sciencedirect, ISI, Google scholar, etc) I agree with the authors claims about the originality of their method. Additionally, the manuscript is generally well-structured and the method is adequately validated. However, there are some minor comments that need to be addressed prior to its final acceptance.

Comments

Line 22: the term of “L” should not be posed as superscript. It is not clear which parameter (peak area or peak height) has been taken into account for the construction of calibration curves. Such information should be added in the text. Figure 2: The authors compared the peak area of the some representative analytes vs the injected sample volume. However, as it can be seen from Figure 1A no baseline-to-baseline resolution has been achieved between mannose & sorbitol and therefore it is questionable the accurate determination of the peak area of each analyte. Table 1: what the authors mean by stating the negative values of time parameter (first column)? Did the authors investigated the robustness of the method?

Author Response

We are grateful to the reviewer for carefully reading the manuscript and for his/her valuable feedback. Please find the points addressed below.

Line 22 – the term of “L” should not be posed as superscript.

Changed as requested.

It is not clear which parameter (peak area or peak height) has been taken into account for the construction of calibration curves. Such information should be added in the text .

Information added as requested.

Figure 2: The authors compared the peak area of the some representative analytes vs the injection sample volume. However, as it can be seen from Figure 1A no baseline-to-baseline resolution has been achieved between mannose & sorbitol and therefore it is questionable the accurate determination of the peak area of each analyte.

We used peak height for quantitation of analytes. The ordinate was mistakenly labelled as peak area and  therefore corrected.

Table 1: what the authors mean by stating the negative values of time parameter (first column)?

The negative values are considered as equilibration time for the column between runs. Equilibration times were evaluated and discussed in chapter 2.1.4. Information added in Table 1 legend.

Did the authors investigated the robustness of the method?

Yes, we investigated the robustness according to ICH guidelines:

The stability of solutions used as mobile phases was evaluated. Microbiological contamination was inhibited using at least 60% ACN in the mobile phases and sample diluents. Samples were diluted not extracted. Therefore, extraction was not evaluated. Influences of pH in the mobile phases were discussed in chapter 2.1.3. Influences in variations in mobile phase composition were discussed in chapter 2.1.1. A huge influence with variation of ACN concentration in the mobile phases can be stated, therefore the volumetric preparation method (v/v) was mentioned in the text. Three different columns (different lots) had been tested with no significant variation in retention times after sufficient equilibration. Temperature and flow rate variations were discussed in chapter 2.1.1 and 2.1.5.

Reviewer 2 Report

The Manuscript reports on validation of the fast, one-run analytical method, based on liquid chromatography (HILIC-CAD), for simultaneous quantitation of various sugars (17 compounds), seven polyols and five common ions with LOQ as low as 0.107-8.918 mg/L and correlation coefficient R2 > 0.97 for all analytes. The method proposed is relatively short, does not require any derivatization steps (no loss of analytes), no complex samples pre-treatment or no use of expensive measuring apparatus such as mass spectrometer coupled with HPLC or NMR spectrometer.

Carbohydrates analysis is controlled in foods and beverages available on the market in Europe by EU laws, which force food producers to meticulous quality control and accurate determination of the nutritional value and composition of their products. Determination of ions, also restricted by EU law, usually requires separate analytical methods such as ion chromatography or ICP-MS. The need for a fast, precise and repeatable analytical method is, therefore, becoming more and more important. The ability to determine many different food components, using one-step analysis and on one apparatus would be undoubtedly an additional advantage. The authors measured all selected compounds (sugars, polyols, and selected ions simultaneously) in 24 different types of popular foods and beverages to confirm the applicability of the proposed analytical method in environmental samples.

Even though, the authors were unable to prevent the coelution of some ion pairs they suggest possible solutions to the problems arose.

I’m honestly impressed with the amount of work authors put into the validation of the proposed method and preparation of the submitted manuscript and therefore I recommend its publication in Molecules without any additional, essential corrections.

Author Response

We cordially thank the reviewer for carefully reading the manuscript and for the positive feedback.